# Prompts for the Future to Live Healthier: A Study of Cognition and Motivation for Healthy Behaviors

**DOI:** 10.3390/ijerph19116682

**Published:** 2022-05-30

**Authors:** Chung-Chih Lin, Pang-Hsiang Yu, Jin-Kwan Lin

**Affiliations:** 1Department of Computer Science and Information Engineering, Chang Gung University, Taoyuan 33302, Taiwan; cclin@mail.cgu.edu.tw; 2College of Management and Design, Ming Chi University of Technology, New Taipei City 24303, Taiwan; jinkwan@mail.mcut.edu.tw

**Keywords:** aging, health belief, unhealthy bedridden time, health promotion

## Abstract

“Aging” is a continuous phenomenon. Medically speaking, physical decline starts after the age of 25. Generally, people do not sense such a decline at a young age, but most transition to some awareness by the age of 50. To enhance the physical and mental health of elderly people and to reduce the length of time spent bedridden, the thoughts and behaviors regarding health and health care among a target group aged between 55 and 75 years were investigated in this study based on the perspective of health beliefs. A total of 300 survey questionnaires were issued and all were returned. The results indicated that after the respondents were reminded of the unhealthy implications of bedridden time, they were willing to enhance their health-promoting behaviors in their diets and regular routines.

## 1. Introduction

In Taiwan, according to statistics from the Ministry of Health and Welfare, the Disability-adjusted Life Years (DALYs) of individuals in the country reached 8.47 years in 2019, which was a new record high over the past eight years. (Along with the issue of Disability-adjusted Life Year (DALY), attention must also be paid to the importance of healthy life expectancy. People find that with the progress of medical treatment, prolonging life is not out of reach, however quality of life is the real goal that should be pursued. As far as impact is concerned, when comparing the average life expectancy of people in different countries, World Health Organization (WHO) no longer only looks at the single figure of average life expectancy, but also looks at the average healthy and unhealthy life expectancy and “the longer average healthy life expectancy and shorter the DALY”, which is regarded as an important indicator of a progressive country). This means that the elderly are unable to live independently in their later years and need to be taken care of. Furthermore, their quality of life is poor half due to chronic disease or disability [1]. Due to medical advances, the average life expectancy is increasing and modern health promotion information is easy to acquire. Why is the number of DALYs still high? In a study, it was found that middle-aged and elderly people are busy with work, therefore they do not pay attention to their health and they have no time to promote health.

In recent years, accidents involving sudden death due to cardiovascular problems or cerebral hemorrhages have become more frequent, and many people may have believed that they were still healthy even though they were very busy. It is easy for people to succumb to chronic conditions such as diabetes, hypertension, and cardiovascular disease after the age of 40. In the early stages of these diseases, one’s situation can be improved through life adjustment. However, after some time, people may need to take medicine to stay healthy in the long run. Therefore, it is necessary to pay attention to one’s health over time in order to reduce the likelihood of becoming bedridden in old age and to improve the quality of life in the longer life expectancy. This study hopes to increase the subjects’ awareness and anticipation of an unhealthy bedridden age by cognitively increasing their own health-promoting behaviors. With regards to changing health-promoting behaviors, increased knowledge through cognitive factors can reduce the excuse of not exercising due to being busy with work. It is important to increase health-promoting behaviors.

The theme of this research is health promotion for middle-aged and elderly people. It is mentioned in the health belief model that changing behavior requires self-benefit. Previous studies have found that when adults are busier with work there is no time for them to engage in health promoting behavior. Awareness of the life expectancy of the subjects and the number of years spent bedridden made them willing to add health-promoting behaviors to their existing health behaviors. The study found that the life expectancy and healthy life expectancy of those who were over the age of 65, retired, and who have a high disposable income were higher. In terms of increasing health-promoting behaviors, having three meals at regular intervals and maintaining a positive mood are items that promote more behavioral improvement. This paper is divided into the following: First, we explain health promotion, the health belief model, and cognitive research on health promotion. Secondly, the research method includes the research object, research design and procedure. Third, a description of some health-promoting behaviors that can be implemented now. Finally, the conclusion and discussion, which illustrates the contribution and limitations of this research in terms of practical suggestions and theory.

### 1.1. Definition of Health Promotion

In 1986, the World Health Organization (WHO) held the first international meeting on health promotion in Ottawa, Canada, and the “Ottawa Charter for Health Promotion” was created. At that time, it was thought that “health promotion” was a process through which people could strengthen their control and enhance their own health. Green and Kreuter (1999) proposed that health promotion be understood as a planned strategy or action through which associations between education, politics, laws, and organizational support promote the healthy life of an individual, group, and community [2]. Generally speaking, health promotion is meant to enhance the health cognition of individuals and groups and guide them toward a correct mindset and aggressive attitude so that their behavior can be changed. Ultimately, health promotion is a way for physical and mental health to be realized to enhance life satisfaction.

Health promotion emerged in the 1990s as a unifying concept which brought together a number of separate, even disparate, fields of study and has become an essential part of contemporary public health [3]. “Health promotion” has been used as a term to describe measures designed to help individuals improve their unhealthy behaviors, or activities designed to encourage positive health [4]. There are many models [5] that frame the scope of health promotion at multiple levels: from global [6], governmental [7], community [8] and individual levels [9,10]. In this study, we focus on the individual level, especially for middle-aged and elderly people. In order to deeply to explore the possibility of behavioral change, we examined the awareness from the perspective of the health belief model.

### 1.2. Health Belief Model

During the early 1950s, the Public Health Service was oriented toward prevention, not the treatment of disease. The health belief model (HBM), influenced by the theories of Kurt Lewin, was developed and includes the motivation and perceptions of an individual. It is useful to explain the problem with health promotion programs [11]. The seriousness of a given health problem may vary from person to person. The degree of seriousness may be judged both by the degree of emotional arousal generated by the thought of disease as well as the difficulties an individual believes a given health condition will create for themselves [12]. Perceived susceptibility and severity have a strong cognitive component that is at least partly dependent on knowledge. There must be at least one action that an individual finds subjectively possible to perform. An alternative action is likely to be seen as beneficial if it relates subjectively to the reduction of one’s susceptibility to, or seriousness of, an illness [11].

The health belief model refers to the actions taken by an individual toward realizing certain health behaviors. It is related to the individual’s self-perception of their possible susceptibility to certain diseases (perceived susceptibility) and their self-perception of the serious consequences of suffering from those diseases (perceived severity). If an individual feels that they could easily acquire a disease and that the result would be very serious, the benefit of adopting certain behaviors to prevent the occurrence of such disease would far outweigh the loss of exploring other behaviors. If there is an appropriate cue stimulus, then the possibility of the individual adopting these behaviors will be greatly increased [13]. After comparing the variables, it was found that a perceived barrier was the most important variable, perceived susceptibility was the next most important to “preventive health behavior,” and perceived benefit was the next most important to “sick-role behavior”, with perceived severity being the least important [14]. Becker and Maiman conceptualize the above process as the HBM [15]. The model is theoretical and has practical applications for the public health field, as seen with COVID-19 [16,17].

The focus of this study was on the use of self-benefit (non-bedridden status) to cross perceived barriers.

### 1.3. Cognition of Health Promotion

Cognition presents a comprehensive review of cognitive psychology which includes sections on representation in memory, abstraction and iconic concepts, symbolic concepts and mental structures, mental operations, consciousness, and search strategies and problem solving [18]. In the HBM, cognition operated in the framework to judge perceived benefits and perceived barriers [12] (p. 333), for example, the belief of one’s ability to care [19], attitude and the affective experience [20], awareness of age and death [21], and knowledge of health and self-efficacy for the intervention [22]. Compare to students in school life are more healthy behaviors associated with healthy literacy [23], it is important to promote healthy lifestyle habits at midlife in order to age in good health [24]. For groups in the age range of 55–75 years, sufficient scientific evidence is needed to motivate them to take specific actions for health promotion. Middle-aged people, due to their career stage, usually do not want to make small-scale behavioral changes. Wen-Wen Shih (2004) surveyed 1778 teachers in colleges and universities and found that most teachers tended to consider work their first priority, which presented a greater barrier to perceived health behavior [25]. This study supported the speculation of health beliefs; however, the public understood the importance of health, but they had less understanding of the relationship between the following three factors: (1) the influence of a future loss of health, (2) the influence of present living mode on quality of life and satisfaction in old age, and (3) the connection between implementing small health behaviors in the present and long-term health and life satisfaction in old age.

## 2. Materials and Methods

### 2.1. Standard for Selection of Research Target

The respondents in this study were 55–75 years old and had a monthly disposable income greater than NT$30,000 (referring to the disposable amount after deducting necessary expenses) or owned real estate. Respondents were recruited via the snowball sampling method (Snowball sampling is a non-probability sampling technique. This questionnaire requires self-disclosure, hence we hoped to find subjects who were willing to answer it. Therefore, the subjects were asked to introduce other subjects to answer this questionnaire through the method of snowballing, which improves self-disclosure and willingness to answer), which is a method of convenience sampling. The final recruited sample size was 300 for the questionnaire survey. All testing procedures were performed in accordance with the 2013 World Medical Association Declaration of Helsinki and were approved by the Research Ethics Committee of Chang Gung Medical Foundation Institutional Review Board in Taiwan (IRB number: 201801878B0C604). The main fieldwork was carried out by 12 trained interviewers between April and May 2020. All respondents provided a phone number, so corrections could be made by phone. This ensured that there were no missing data in the end.

### 2.2. Research Procedure

The survey questionnaire was designed based on secondary data collection and the anticipation of health behaviors. To prepare the draft of the quantified survey questionnaire, the questions were developed from the 2D factor dimensionality theory. The questions were then revised after expert discussion and pre-test procedures to settle on a final version. The survey questionnaire was then reviewed, revised, and adjusted in terms of its guide words by the Chang Gung Medical Foundation Institutional Review Board. The survey questionnaire was issued to targets conforming to case-receiving conditions; in the end, the number of returned survey questionnaires that were considered effective was 300.

The content of the survey questionnaire included the following parts:Social demography variables, including age, gender, present work situation, occupation, educational level, etc.Perceived health conditions: This was a self-report of the sample on their health status, including perceived health conditions, health level as compared to that of 3 years ago, prediction of self-health, and autonomous non-bedridden age.Living habit situation: After reminding the respondents of what unhealthy bedridden years were, the sample showed a stronger willingness to continue with behaviors that he/she had implemented, or increased behaviors that could be implemented presently. The contents included previous dieting methods, present dieting methods, previous physical activity, and present physical activity.

In total, there were 57 items, including 10 demographic items.

## 3. Results

Among those who responded to the questionnaire, 52% were female, and age was mostly in the range of 55–64 years old (60%). Regarding religion, the highest percentage (41%) were Buddhist. For educational level, those above senior high/occupational school comprised 77% of the sample. With respect to birthplace, Islanders and Southern Fujian comprised about 71%. A total of 56% were retired, and about 30% worked in commerce. For marital situation, married or cohabitating individuals comprised about 82% of the sample, and 78% were living with a partner/spouse. A total of 70% had experience taking care of parents. The demographic information with the largest proportion of personal basic information for the 300 questionnaire subjects is shown in Table 1.

The main results found from the questionnaire survey analysis are as follows:(1)Self-health situation

Those with a good perceived health situation (“very good” and “good”) comprised 54%, 42% perceived their health situations as fair, and 4% reported bad health situations. Those under the age of 64 who were presently employed with a higher disposable income tended to have a better health situation. The arrow in Figure 1 means those with higher disposable income have a relatively higher proportion of better health.

(2)Self-age prediction

Of those in the sample, 49% felt that they might be able to live up to 81–90 years old and 12% felt that they might live beyond 91 years old. As the Figure 2 shows, the average expected age was predicted to be 84.9 years. Three sub-groups, consisting of those presently above 65 years old, those who were retired, and those with higher disposable incomes, generally felt that they would live longer.

(3)Prediction of age of self-health and autonomous non-bedridden age

Of the respondents, 59% felt that they could not have a healthy, autonomous, and non-bedridden life before 80 years old, 36% felt that they could still have a non-bedridden life before 81–90 years old, and 6% felt that they could have an autonomous and non-bedridden life even after 91 years old. The average age at which individuals predicted that they would still be autonomous and non-bedridden was 81.5 years. Those above 65 years old who were retired generally felt that they would have autonomous and non-bedridden lives at a higher age. The detail of the subgroups, please refer the Figure 3.

(4)Health situation compared to that of three years ago

Compared to their circumstances three years ago, the differences in the respondents’ self-evaluated health situations were reported as follows “memory was worse than before” (69%), “physical strength was worse than before” (59%), “did not know the cause of waist and back pain” (31%), “red numbers appeared in health report” (27%), and “catching a cold more frequently” (7%). Only 8% did not have any of the above issues. Those who were still employed reported not knowing the cause of waist and back pain at a higher rate. People with a monthly disposable income of more than NT$50,000 had a higher tendency to report that their memory was worse than before. Those with a monthly disposable income of less than NT$30,000 reported not knowing the cause of their waist and back pain at a higher rate (see Table 2 for details).

(5)Previous dieting method

The dieting methods reported to have previously been adhered to were mainly and sequentially “less or reduced sugar in the diet” (76%), “fixed time and quantity in three daily meals” (72%), “having two kinds of fruits every day” (58%), “having vitamins or other health food every day” (53%), and “having three dishes of vegetables every day” (51%). Less than half of the subjects had the habit of “eating food containing husked rice or the five cereals”. Meanwhile, subjects above 65 years of age were more likely to eat two fruits and vitamins or other health foods every day, compared to those below 64 years of age. Moreover, a relatively higher percentage of retired people ate a fixed quantity at specific times for three meals each day, with two fruits and three dishes of vegetables each day.

(6)Previous physical activity

Previous physical activity included, in sequential order, “maintaining a positive mood” (83%), “sleeping more than six hours daily” (79%), “maintaining autonomous exercise habits” (54%), “sitting for less than 90 min at one time each day” (43%), and “walking 6000 steps daily” (40%). Less than 30% had attended a sports club. Relatively speaking, people above 65 years old tended to have more autonomous habits than people below 64 years of age. They also tended to be less likely to sit for a duration of more than 90 min than people below 64 years of age. Those who were retired tended to have more autonomous exercise habits, were more likely to attend sports clubs, and were less likely to sit for a duration of more than 90 min. Meanwhile, people with monthly disposable incomes of less than NT$30,000 tended to be less likely to participate in sports and fewer have autonomous exercise habits (see Table 3 for details).

Sub-summary: After reminding the respondents aged 55–75 years of age of the significance of unhealthy bedridden years, “dieting methods that can be adopted now” and “physical activities that can be done now” were respectively determined and are shown in Table 3 and Table 4.

(1)Present dieting methods

For the sake of their health, dieting methods presently employed were mainly and sequentially as follows: “having fixed times and quantities for three meals each day” (86%), “having less or reduced sugar” (85%), “having two fruits each day” (80%), “having three dishes of vegetables daily” (73%), “eating as much husked rice and cereals as possible” (65%), and “taking daily vitamins or other health foods” (63%). The number of people who were willing to watch their diets after completing the questionnaire survey is shown in Table 4.

(2)Present physical activity

For the sake of health, present physical activity were sequentially as follows: “maintain a positive mood” (89%), “sleeping for more than six hours daily” (85%), “having autonomous exercise habits” (72%), “walking 6000 steps each day” (62%), and “sitting for less than 90 min each time” (60%). The number of people who were willing to engage in more physical activity after finishing this survey questionnaire is shown in Table 5.

## 4. Discussion

Among the 300 responses to the quantitative survey questionnaires, it was found that most of the participants thought that they could not have a healthy, autonomous, and non-bedridden life before 80 years old, even if dieting adjustments were made and physical activity levels were increased at the present time. Although body function deteriorates gradually, leading to the occurrence of more accidents, participants were still worried that they were insufficiently managing their health at the present time. Meanwhile, after reminding those of the significance of unhealthy bedridden years, in addition to the health behaviors that had been engaged in previously, respondents were willing to add more health promotion behaviors to their diets and daily routines in order to avoid the occurrence of bedridden situations. For this study we used convenience sampling and trained interviewers to collect the data. The results are not obtained from standard sampling procedures, so the power of the generalization is limited. The implication of these results should be mentioned in a conservative manner. In the future, studies should use more standardized sampling procedures.

## 5. Conclusions

To avoid a bedridden life, it is necessary to change bad living habits and to have periodical health checks. Because diseases can accumulate from small actions, prevention is better than a cure. If people have little knowledge about the future, they tend to make no changes; however, after they learn about something that they might have to face in the future, they may be more likely to change their present situation. This may also reduce unnecessary expenses in national healthcare systems. Ultimately, the goal is for people to enjoy a healthy and happy life.

## Figures and Tables

**Figure 1 ijerph-19-06682-f001:**
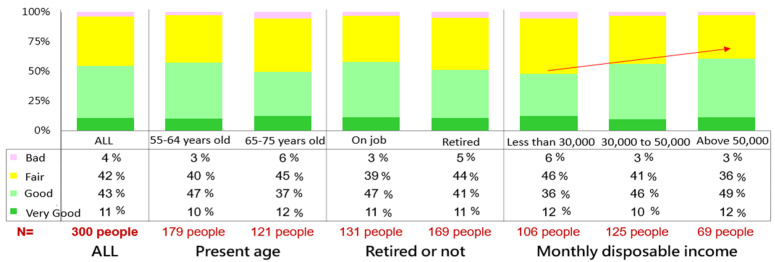
The self-health situation of those aged 55–75 years.

**Figure 2 ijerph-19-06682-f002:**
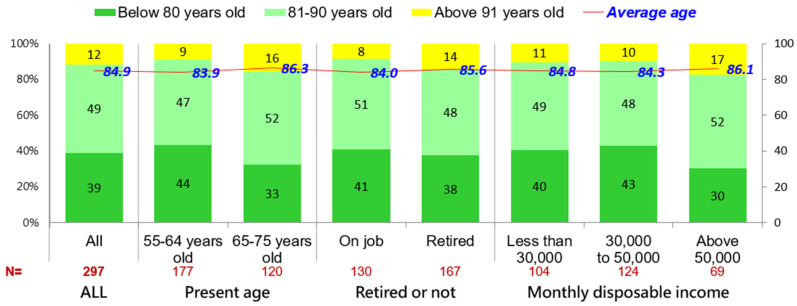
Self-age prediction of those aged 55–75 years.

**Figure 3 ijerph-19-06682-f003:**
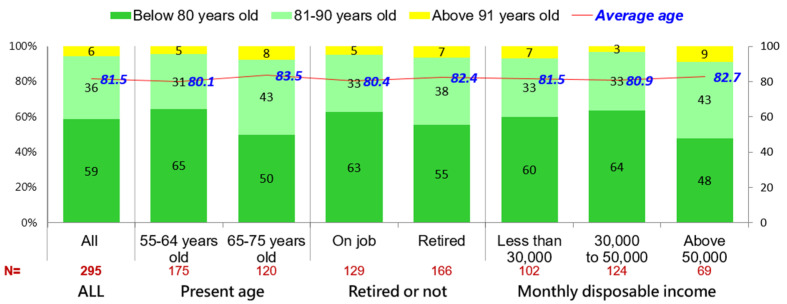
Predicted age of self-health and autonomous non-bedridden life for those aged 55–75 years old.

**Table 1 ijerph-19-06682-t001:** Profile information of the sample.

Basic Information	Top Item	Percentage
gender	female	52%
age	in the range of 55–64 years old	60%
religion	Buddhism	41%
educational level	those above senior high/occupational senior high school	77%
birth place	Islanders and Southern Fujian	71%
occupational situation	retired	56%
occupation	commerce section	30%
marital situation	married or cohabiting	82%
present living situation	living with partner/spouse	78%
experience taking care of parents	had experience taking care of parents	70%

**Table 2 ijerph-19-06682-t002:** Self-health situation for those aged 55–75, as compared to their self-health situation three years ago.

%	All	Age		Retired or Not		Monthly Disposable Income		
		55–64 Years Old	65–75 Years Old	On Job	Retired	Less Than 30,000 NTD	30,000–50,000 NTD	Above 50,000 NTD
*n* =	300	179	121	131	169	106	125	69
Memory was worse than before	69%	69%	70%	71%	68%	68%	66%	78%
Physical strength was worse than before	59%	60%	59%	62%	57%	60%	59%	58%
Not knowing the reason for waist and back pain	31%	35%	26%	37%	26%	35%	31%	25%
Red letter shown in health test report	27%	28%	25%	30%	24%	28%	27%	23%
Catching cold more	7%	7%	7%	6%	8%	6%	10%	6%
frequently	8%	8%	7%	8%	8%	8%	7%	9%

**Table 3 ijerph-19-06682-t003:** Physical activities previously done by those aged 55–75 years.

%	All	Age		Retired or Not		Monthly Disposable Income		
		55–64 Years Old	65–75 Years Old	Employed	Retired	Less Than 30,000 NTD	30,000–50,000 NTD	Above 50,000 NTD
*n* =	300	179	121	131	169	106	125	69
Maintaining a positive mood	83%	79%	88%	76%	88%	80%	83%	86%
Sleeping more than six hours each day	79%	81%	77%	79%	79%	79%	79%	80%
Having autonomous exercise habits	54%	50%	60%	47%	59%	45%	62%	52%
Sitting less than 90 min at a time	43%	38%	50%	37%	47%	41%	46%	42%
Waling more than 6000 steps a day	40%	36%	45%	38%	41%	39%	41%	41%
Participating in a sports club	27%	23%	31%	21%	31%	25%	26%	30%

**Table 4 ijerph-19-06682-t004:** Comparison of present dieting methods.

Item	Dieting Methods That Were Done Previously	Dieting Methods Done Presently	Number of People Who Were Willing to Pay More Attention to Diet after This Survey	Percentage
Try to have fixed times and quantities for three daily meals	72 persons	86 persons	+14 persons	29%
Try to have less or reduced sugar	76 persons	85 persons	+9 persons	28%
Have two types of fruits daily	58 persons	80 persons	+22 persons	27%
Have three dishes of vegetable daily	51 persons	73 persons	+22 persons	24%
Try to eat a lot of husked rice and cereals	46 persons	65 persons	+19 persons	22%
Take daily vitamins or other health foods	53 persons	63 persons	+10 persons	21%

**Table 5 ijerph-19-06682-t005:** Comparison of physical activity done previously and presently.

Item	Physical Activities That Were Done Previously	Physical Activities That Were Done Presently	Number of People Who Were Willing to Engage in More Physical Activity after This Survey	Percentage
Maintaining a positive mood	83 persons	89 persons	+6 persons	30%
Sleeping for more than six hours each day	79 persons	85 persons	+6 persons	28%
Having autonomous exercise habits	54 persons	72 persons	+18 persons	24%
Walking 6000 steps each day	40 persons	62 persons	+22 persons	21%
Sitting for less than 90 min at a time	43 persons	60 persons	+17 persons	20%
Participating in a sports club	27 persons	37 persons	+10 persons	12%

## Data Availability

Data can be acquired from the correspondence author according to reasonable request.

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
