# Peer review of "Prompts for the Future to Live Healthier: A Study of Cognition and Motivation for Healthy Behaviors"

_ijerph, 2022, doi:10.3390/ijerph19116682_

Round 1

Reviewer 1 Report

We think this is an interesting report that examines the actual state of healthy life expectancy and efforts to extend it for the elderly in Taiwan. In the discussion, it is mentioned that there are differences in awareness depending on whether they are employed, not employed, or have high or low income.

Line 24: I understand the meaning of "unhealthy life expectancy," but I think the phrase "healthy life expectancy" is generally used. Please explain why you chose to use the term unhealthy life expectancy.
Line 37-40: Please clarify the purpose of this study.
In particular, the question to the reader, "However, can busy work be removed by cognition factor?" is not a scientific statement.

Reviewer 2 Report

This could be an interesting article for the readers of this journal. In the same way, I hope the following comments may be helpful to improve the quality of the paper:

  1. The paper must pass proofreading.
  2. The Introduction must be fully developed (i.e, presentation of the topic, background and summarizing of existing research on this topic, theoretical perspective and research approach adopted in this paper, research problem, main conclusions, and structure of the paper).
  3. The structure of the paper must correspond accordingly.
  4. Authors should ensure that they have done a thorough review of the literature.
  5. Authors should discuss in more detail and the methodological approach adopted in this paper.
  6. Conclusions must be improved and developed accordingly to the results achieved in the paper.

Round 2

Reviewer 1 Report

The authors have appropriately revised it according to the reviewer's comment. This reviewer thinks this manuscript could be accepted for publication.

Author Response

Dear Reviewers and Editors,

Thank you for you despite your busy schedules, have taken out time to read our manuscript and give us the valuable comments and critical review which for sure improved the clarity and quality of our manuscript. In these two revisions, we modify these points:

  1. Strengthens the introduction, on 1.1 Definition of health promotion, 1.2 Health belief mode, and 1.3 A study of the cognition of health promotion. And add 21 reference articles.
  2. Clarify the methodology, especially on the snowball way, the trained interviewers and procedures of the data collection.
  3. Detailed the content of the table 2 and table 3, add the mark of the percentage, “%”.
  4. Proofread from the native professional editor of the Scribendi English editing service company.   
  5. And other modified to response the reviewers’ suggestions.

We really appreciate your effort in this manuscript. And we have learned how to be more professional and clarity to present our thoughts. Thank you!

Best Regards,

Chung-Chih Lin

Pang-Hsiang Yu*

Jin-Kwan Lin

Reviewer 2 Report

  1. The Introduction has improved. However, the subsections 1.1 Definition of health promotion, 1.2 Health belief mode, and 1.3 A study of the cognition of health promotion should be included as part of a (new) different section.
  2. In this regard, authors must be sure that all relevant literature (literature review) has been included and updated to support this (new) section.
  3. The discussion of the research design should be improved (2. Materials and Methods).

I hope you will find these comments supportive to improve this paper. Thank you.

Author Response

(The authors gave the same response as above.)

Reviewer 3 Report

Dear Authors,

The manuscript looks and reads much better than the previous submission. My comments/suggestions were adequately answered.  

Author Response

(The authors gave the same response as above.)
